# Crowd Monitoring in Smart Destinations Based on GDPR-Ready Opportunistic RF Scanning and Classification of WiFi Devices to Identify and Classify Visitors' Origins

Alberto Berenguer [1] , David Fernández Ros [2], Andrea Gómez-Oliva [2], Josep A. Ivars-Baidal [3], Antonio J. Jara [2,*] , Jaime Laborda [4] , Jose-Norberto Mazón [1] and Angel Perles [4]

1. Instituto Universitario de Investigación Informática, Universidad de Alicante, 03690 San Vicente del Raspeig, Spain; aberenguer@dlsi.ua.es (A.B.); jnmazon@ua.es (J.-N.M.)
2. HOP Ubiquitous S.L. (HOPU Smart Cities), Luis Buñuel. 6, 30562 Murcia, Spain; davidfr@hopu.org (D.F.R.); andrea@hopu.org (A.G.-O.)
3. Instituto Universitario de Investigaciones Turísticas, Universidad de Alicante, 03690 San Vicente del Raspeig, Spain; josep.ivars@ua.es
4. ITACA Institute, Universitat Politècnica de València, Camino de Vera s/n, 46022 Valencia, Spain; jlaborda@itaca.upv.es (J.L.); aperles@disca.upv.es (A.P.)
* Correspondence: jara@ieee.org

**Abstract:** Crowd monitoring was an essential measure to deal with over-tourism problems in urban destinations in the pre-COVID era. It will play a crucial role in the pandemic scenario when restarting tourism and making destinations safer. Notably, a Destination Management Organisation (DMO) of a smart destination needs to deploy a technological layer for crowd monitoring that allows data gathering in order to count visitors and distinguish them from residents. The correct identification of visitors versus residents by a DMO, while privacy rights (e.g., Regulation EU 2016/679, also known as GDPR) are ensured, is an ongoing problem that has not been fully solved. In this paper, we describe a novel approach to gathering crowd data by processing (i) massive scanning of WiFi access points of the smart destination to find SSIDs (Service Set Identifier), as well as (ii) the exposed Preferred Network List (PNL) containing the SSIDs of WiFi access points to which WiFi-enabled mobile devices are likely to connect. These data enable us to provide the number of visitors and residents of a crowd at a given point of interest of a tourism destination. A pilot study has been conducted in the city of Alcoi (Spain), comparing data from our approach with data provided by manually filled surveys from the Alcoi Tourist Info office, with an average accuracy of 83%, thus showing the feasibility of our policy to enrich the information system of a smart destination.

**Keywords:** smart destination; GDPR; crowd monitoring; WiFi scanning; people counting; IoT; FIWARE; RF scanning; COVID-19; Smart Cities

## 1. Introduction

The smart destinations concept was coined as a distinct step in the evolutionary relationship of ICT (Information and Communications Technologies) and tourism, characterised by integrating the physical and the digital world [1]. Innovative ecosystems have emerged as tourism systems that take advantage of novel technologies (such as Internet of Things, IoT) and the intensive use of information in creating, managing and delivering intelligent tourist services/experiences [2]. A new scenario appears under the smart destination approach [3], contributing to improving sustainability, competitiveness and tourist experiences [4].

The generation of new information systems is one of the main pillars of smart destinations. These systems include market intelligence linked to the different phases of the travel life cycle (inspiration, booking, experiencing and sharing) and the monitoring of tourist behaviour at the destination as crucial information for advanced management.

The Destination Management Organisation (DMO) should function as a smart hub that coordinates all relevant sources of information and makes it accessible to different users [5]. In the case of tourist behaviour monitoring, the DMO must deploy a technology layer that allows data gathering for other purposes.

In this context, measuring the spatial behaviour of tourism is a challenge for both researchers and practitioners, above all tracking visitor movements and their concentration at particular points of interest [6]. Crowd monitoring was an essential measure to deal with over-tourism problems in urban destinations in the pre-COVID era [7]. Likewise, it will play a key role in the pandemic scenario when restarting tourism and making destinations safer [8]. Therefore, the DMO must have enough and high-quality crowd monitoring data to make informed decisions and offer visitors a secure experience while keeping residents safe, e.g., avoiding the overcrowding of visitors.

However, crowd monitoring by the DMO has not been widely adopted due to the need to associate people with some tracking technologies while complying with the privacy restrictions imposed by regional legislation. It should be noted that this work is geographically framed within the European Community, so Regulation (EU) 2016/679 [9] applies. This regulation is better known as the General Data Protection Regulation (GDPR) and seeks to harmonise data privacy laws across Europe. This is one of the most restrictive legislations in the world in terms of safeguarding citizens' privacy. Therefore, it is a not easy for an effective European DMO to comply with such a regulation, quite the contrary, as most efforts in crowd monitoring have been made by using image processing, hindering the enforcement of privacy regulations [10].

On the other hand, mobile phone operators and companies such as Google and Apple know all our movements in real time (and, in many cases, our interests). Paradoxically, the exploitation of this information is forbidden by the DMO in order to comply with the GDPR. Moreover, it is also essential for the DMO to avoid total dependence on data providers such as telecommunication companies or booking platforms. Despite restrictions on widespread access to information collected by mobile phones, DMOs have devised ways to circumvent legal limits by promoting user consent, directly or indirectly, to share information. The most commonly used approaches are destination apps, QR (Quick Response) codes and NFC (Near-Field Communication) tags [3,6]. Unfortunately, it is challenging to convince tourists to install destination apps, and they are unlikely to use the QR or NFC tags made available by tourism services.

WiFi-based approaches have emerged lately as a promising way to perform crowd monitoring to overcome these shortcomings, as they better fit with privacy regulations [10]. WiFi scanning of mobile phones with the WiFi interface enabled can capture the unique MAC (Media Access Control) address of the devices to track and only count the number of people in a specific space to avoid incurring privacy issues. A prominent open project in this regard is [11]. In this sense, techniques for estimating the number of mobile devices present at a particular location and time by analysing WiFi probe requests from smart devices have been proposed so far (e.g., ref. [12]). Interestingly, other information also exposed by WiFi-enabled mobile devices via probe requests is the list of preferred WiFi access points (the Preferred Network List, aka PNL) in the form of SSIDs (Service Set Identifier).

The research question that guides this work is: Would it be feasible to exploit this WiFi scanning information to go beyond detecting the number of persons at a certain location and time, and distinguishing visitors from residents in a smart tourism destination while, of course, remaining compliant with the GDPR? As stated by [13], this is still considered an open problem for smart tourism destinations, and it is also an initial step to further consider tourist digital footprints or data traces from tourist activities (as they occur if a person can be considered a tourist) [13].

To this end, the main contribution of this paper is an approach to (i) gathering crowd data by processing smartphone device signals based on WiFi scanning, as well as (ii) differentiating the percentage of visitors and residents in a smart tourism destination. Our approach is based on the mass collection of local SSIDs and comparison with the preferred

WiFi network information exposed by mobile terminals. In order to comply with GDPR, encryption and data locality mechanisms have been devised to ensure privacy. A pilot study has been conducted in the city of Alcoi (Spain) comparing data from our approach with data provided by manually filled surveys from the Alcoi Tourist Info office. Initial results show that our approach is a viable solution that could help the DMO to make informed decisions and offer visitors a secure experience, while also keeping residents safe, avoiding overcrowding in the pandemic scenario.

At this point, it should be noted that our approach complements other types of initiatives from DMOs based on the purchase of data from third parties (mobile providers). In fact, many DMOs have based their decision-making process on data acquisition from mobile providers, but this attempts against independence since it is difficult to ensure high-standard data governance principles. Moreover, relying solely on third party data means that DMOs are constrained by budgetary availability.

The article is structured as follows: After briefly describing related work in Section 2, Section 3 presents our crowd monitoring approach for detecting visitors and residents of a smart tourism destination. Section 4 shows the results of a pilot study in the city of Alcoi (Spain). Finally, the discussion about the obtained and future work is presented in Section 5.

## 2. Related Work

In recent years, the study of crowd monitoring has undoubtedly been growing. Traditionally, one of the most widely used approaches has been computer vision; however, as pointed out by [10], due to predominant use of videos/image sequences, the existing techniques may raise data privacy concerns. The advent of sensors capable of recognising WiFi signal from smart devices has brought several improvements, such as the possibility of covering a larger area without the need for visual contact, but also the possibility of development novel crowd monitoring techniques which are privacy-preserving and require minimum human participation.

There are many crowd monitoring approaches based on WiFi scanning. For example, ref. [14] analyses the flow of people in airports and its comparison with ground truth provided by the security check process in order to discuss the quality and feasibility of pedestrian flow estimations for both WiFi and Bluetooth scanning. Their conclusions encourage the use of WiFi scanning for crowd monitoring. On the other hand, the authors in [15] propose the use of the list of preferred SSIDs of detected smart devices in urban areas for crowd monitoring. They establish and maintain an SSID database by integrating both offline and online information. However, this proposal does not take into account privacy restrictions imposed by regional legislation.

In [10], an analysis of the state of the art is performed, where the different problems that crowd monitoring can solve are established, such as measuring the duration of queues, estimating trajectories and social relations, estimating population density in a specific area, etc. Importantly, all of this must always be carried out in compliance with current privacy legislation.

If we focus on the field of tourism, several crowd monitoring proposals have been developed with the aim of detecting visitors with special emphasis on trying to determine how they move around the tourist destination. Some of them are briefly described below.

For example, in [13], so-called passive mobile data (i.e., event data recorded by mobile network operators in the course of mobile phone usage by a consumer connected to public voice and data networks) is used to track visitors in a tourist destination. The advantage of using these data is that they can be recorded on the network without any activity on the part of the user: it is automatically generated as soon as a mobile device, the communications tower and the phone network operator's system communicate. However, the main drawback is that these data are the property of the telecommunication companies, so if the tourist destination needs to acquire this information, it is not always feasible due

to budget constraints. Additionally, independence of tourist destinations is lost, and data governance becomes more complex.

Other proposals, such as the one developed in [16], make use of other technologies such as NFC (Near-Field Communication), which is a wireless technology for data transfer without physical contact. Specifically, ref. [16] studies how to apply NFC in the tourism area, emphasising the opportunities but also analysing the threats and problems of its use. In particular, it can be highlighted that NFC has a short range, which limits its use to specific situations where the person using it carries out an activity (e.g., payment or use of public transport).

There are other works in the tourism area, such as that described in [17], which makes use of a different type of data, namely bank transaction information. In this case, the aim is to complement official statistics based on surveys to make up for this lack of reliable information. These statistics are complemented by the use of card transaction data (both payments and ATM cash withdrawals). The underlying idea is to detect patterns in banking transactions in order to determine whether the person making the transaction is a visitor or a resident. However, the problem remains similar to the use of data from telecommunication companies as aforementioned, as bank transaction data belong to companies and, in order to use them, they need to be purchased.

One work that uses data from WiFi signals is [18], where a method is presented to track people at mass events in tourist cities without the need for the use of a mobile application or QR codes or similar mechanisms that require people's active participation. This proposal is based on scanning at various locations the packages sent by the WiFi interface of visitors' smartphones, and correlating the data captured at these different locations. However, privacy concerns are not discussed in detail in this approach.

Lately, novel approaches have emerged for detecting people and understanding their behaviour automatically [19]. For example, some approaches (such as [20]) attempt to transform an unmodified WiFi radio infrastructure into a flexible sensing system for detecting the people moving indoors. Other proposals, such as [21], develop their own sensor nodes to count the number of pedestrians and their direction of travel along with some ambient parameters. Interestingly, proposals such as [22] use a stereo thermal camera setup for pedestrian counting and behaviour understanding regardless of light conditions. Finally, the unexpected COVID-19 pandemic scenario has encouraged the development of systems monitoring the achievement of social distancing. In this sense, ref. [23] proposes using low-cost Raspberry Pi and a variety of sensors to measure temperature, distances, etc., in order to ensure the COVID-19 standard operating procedure compliance.

Finally, the work developed in this article is designed to be used to obtain the number of visitors to a tourist destination (it is not intended to track them, only to determine whether they are visitors or residents) while remaining compliant with current legislation and trying to overcome some of the aforementioned pitfalls. Specifically, our proposal (i) does not require active user participation; (ii) does not require buying data from third parties such as banks or telecommunications companies (i.e., the destination can have more simple data governance); (iii) does not have a limited scope; and (iv) does not depend on the use of a service (such as public transport) to acquire data.

## 3. Visitor Crowd Monitoring Approach

### 3.1. Architecture

Our approach aims to collect and process data from mobile devices to estimate the number of people visiting a destination (i.e., visitors) relative to the number of residents for each day of the year. For this purpose, the proposed visitor crowd monitoring approach takes advantage of smartphone devices with an enabled WiFi interface to periodically scan nearby WiFi access points available for connection [12]. This is completed by sending (regardless of whether the device is connected to a WiFi access point) control frames named "probe requests". Information in this probe request contains, among others, the Preferred Network List, which includes the identifier of the WiFi access points (SSID) to

which the device has already connected [12]. This way, our visitor crowd monitoring approach identifies the mobile devices whose WiFi SSID stored as preferred for connection is located in the destination (i.e., residents) and differentiates them from those devices whose preferred SSID is not from the destination (i.e., visitors).

Figure 1 shows a diagram of the proposed architecture, whose main elements are described below:

- A collector of SSIDs of the existing WiFi access points in a specific location (e.g., a city). This collector aims to create an accurate database of the available SSIDs in a tourist destination (top left of the figure).
- A collector of SSIDs of preferred WiFi access points coming from probe request data frames of mobile devices detected by HOPU Smart Spots [24] (top right of the figure).
- A postprocessing cloud infrastructure to determine the daily number of visitors in different locations around the destination (bottom of the figure).

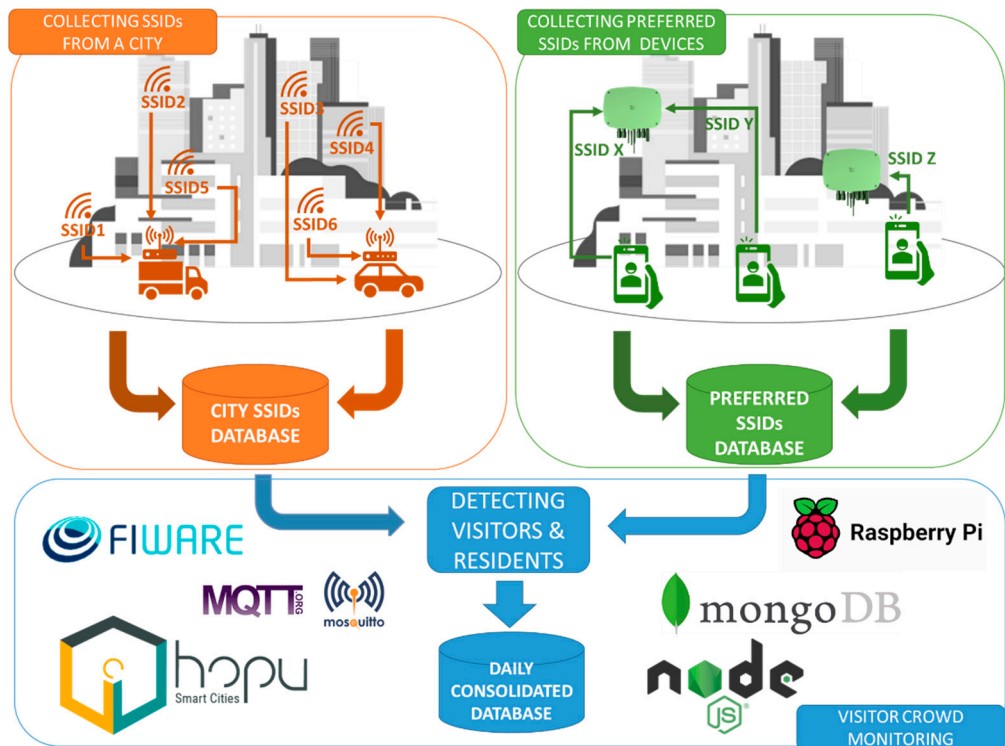

**Figure 1.** The architecture of our visitor crowd monitoring approach.

In the following subsections, each of the parts of the proposed approach are described in detail.

### 3.2. Collection of SSIDs from a Smart Destination

WiFi access points usually expose their SSIDs by using periodic beacon frames. This is a regular advertisement to inform any listening devices that this SSID is available and has particular features. Client devices depend upon these beacon frames to discover which networks are available (passive scanning) and to ensure that the networks that they are associated with are still present and available.

These SSIDs can be 0 to 32 bytes long and are, in general, a natural language string of random characters. An analysis of the exposed patterns worldwide shows that these strings tend to be locally identifiable and unique.

Collecting SSIDs is fairly immediate and has been massively employed by companies such as Google and Mozilla, among others, to improve their mobile device geolocation systems. A fairly comprehensive table of companies that collect this information for the so-

named WPS (WiFi Positioning System) and its availability can be found in [25]. Capturing SSIDs is not exempt from controversy, with a notable scandal around the surreptitious gathering of WiFi data while capturing video footage and mapping data for Google's Street View service [26]. As a result of this scandal, proposals such as adding the tag "opt_out" to the SSIDs have been applied so that users can opt out of the capture of this service.

The legality of whether or not this information can be collected, especially in the European space, is unclear. In our particular case, the project is part of municipal public service (city of Alcoi in Spain), so extreme measures have been taken in terms of legality and privacy, in agreement with the legal advisory services of the municipality. The approach followed immediately encrypts each of the captured SSIDs and stores only the timestamps and the city where it was caught (no more specific location is required).

Although the most immediate implementation of the SSID capture system would be using a mobile phone application such as WiGLE [27], the decision was taken to develop a custom SSID collector to simplify the development and improve the performance of SSID frame capture using high gain antennas. WiGLE (or Wireless Geographic Logging Engine) is a website for collecting information about the different wireless hotspots around the world. Users can register on the website and automatically upload hotspot data such as GPS coordinates, SSID, MAC address and the encryption type used on the hotspots discovered using a mobile application. Considering that this information is open, we decided to use it as a reference to compare some of the data collected by WiGLE and the one collected by our system.

To validate the proposal, the following setup was used:

- A Smart Spot by HOPU with a ESP32 chipset, alternatively it can be used a Raspberry Pi 4 Model B computer. This is a popular low-cost computer capable of running a Linux operating system, thus allowing complex applications.
- A dual-band USB WiFi dongle with high-gain antenna model TP-Link Archer T3U Plus with a sensitivity of about $-75$ dBm for 2.4 GHz band and $-70$ dBm for 5 GHz band.

HOPU Smart Spot IoT are environment monitors based on IoT technology enabling to monitor noise, people affluence/density and gases: nitrous vapours ($NO/NO_2$), sulphides ($H_2S/SO_2$), carbon monoxide/dioxide ($CO/CO_2$), Ozone, and other toxic substances as alcohols, ouds (VOCs); particulate matter (PM) to identify specific nanoparticulate such as dust (PM10), pollens (over PM40), pollutants (PM2.5) and viruses (under PM1). Being able to monitor levels of toxicity, pollution and air quality; with high reliability and data quality (certified and verified by our own certified labs).

It is remarkable that the solution presents high reliability and data quality. HOPU has a sophisticated high precision lab, where through Machine Learning algorithms we improve the precision of our sensor measures reducing the effect of cross-sensitivity. Air quality sensors are calibrated and validated in a laboratory with high accuracy and certified reference data.

Smart Spot is an all-in-one Smart Cities and Smart Destination contextual information collector. Regarding crowd-monitoring, Smart Spot antenna type chosen for the SSIDs scanner has been a key element, as we intend to capture SSIDs of adjacent buildings from the ground level. In this sense, we selected an omnidirectional antenna with a gain of about 5 dBi to offer a reasonable compromise between range and a good reception pattern to pick up the SSIDs of the WiFi routers installed in the adjacent buildings.

Smart Spot collects SSID data from WiFi networks in a city and the SHA1 hash algorithm is applied to the SSID of any WiFi access point encountered, as well as the date of capture in timestamp format and the name of the city where it occurred.

Specifically, we have developed two software components:

- The first is focused on collecting all the available SSIDs nearby. It aims to select the network interface used to capture the data and set a refreshing time (the time between the capture of each SSID). Once configured, this software runs a script that returns the available SSIDs and then cleans the output and applies an SHA1 hash encoding to ensure the privacy of the data. Finally, the data collected are written into a flat file in

an SD memory. This software is running as a service, which means that if the device is on and the wireless card is connected, it will be collecting data.

- The second part takes care of exporting the data from the Smart Spot to a cloud database described later. Database availability is checked before stopping the service from running the SSID collection. It sets the date and the location, and then reads the file generated by the other script, thus inserting each SSID saved with the current date and the location. Finally, it checks if the number of uploaded files corresponds to the number of SSIDs which appear on the text file to ensure that all the data collected are exported to the server. If everything is correct, the text file is deleted, and the service of the collection script is restarted.

### 3.3. Collection of Preferred SSIDs from Mobile Devices

The collection of preferred SSIDs is conducted by a modified version of the HOPU Smart Spot device shown in Figure 2. This is a multi-sensor IoT device devoted to smart-city applications. In our case, we benefit from its WiFi crowd monitoring capability, which has been suitably adapted to achieve the desired characteristics described below.

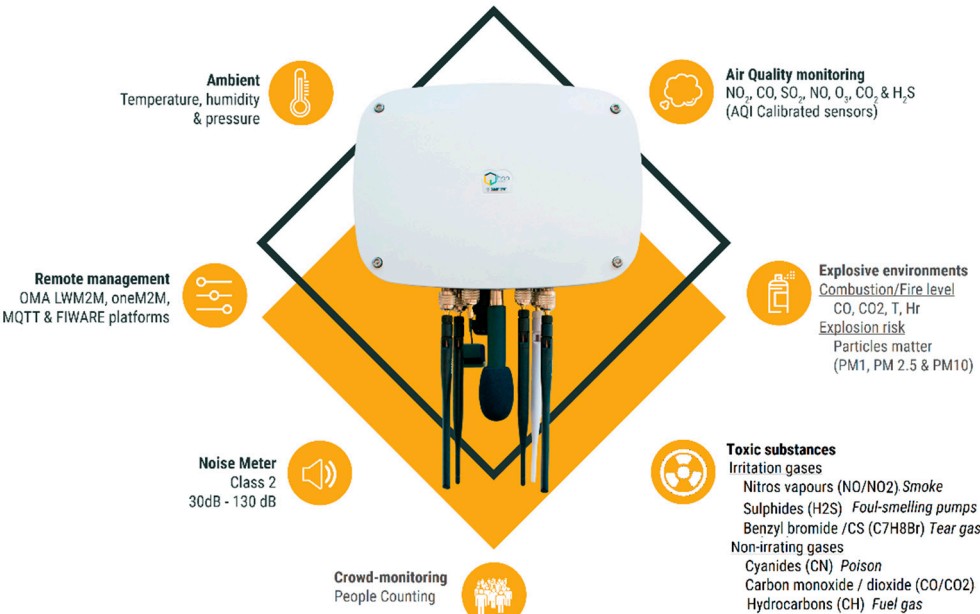

**Figure 2.** HOPU Smart Spot IoT device (HOPU Smart Spot IoT device offers Open Source libraries and framework for research purposes, smart cities and smart destination projects. Please contact: jara@hopu.org for your enqueries and check for more details: https://hopu.eu/ (accessed on 12 January 2021)).

The HOPU Smart Spot device listens for "probe request" packets sent by mobile devices. These packets correspond to those sent by mobile phones when searching for a WiFi access point to connect to. Normally the search for a WiFi access point may be accompanied by the name of the particular WiFi access point being searched for. Once a mobile device has been detected, within HOPU Smart Spot itself, an SHA1 hash algorithm is applied to the MAC and SSID (from the PNL) detected, making it impossible to obtain the MAC and SSID back again, thus anonymising the user's information on the device itself after it has been received.

MACs and SSIDs anonymised using the SHA1 hash are stored in volatile memory and counted in 1, 5, and 10 min intervals internally within the device. MACs (and corresponding SSIDs from the PNL) are discarded/deleted from memory when they are not detected by the device and are outside the maximum time range (10 min).

Every minute, the HOPU Smart Spot device counts the devices detected in each of the intervals (1, 5, and 10 min) and sends this information through the integration protocols available in the device (OMA LwM2M, MQTT, LoRa, SENTILO and FIWARE NGSI).

This whole process of hashing MACs and SSIDs within the device itself and not storing them in non-volatile memory is carried out in order to protect citizens' data in accordance with the current data GDPR protection law.

It should be noted that the SSID collection is performed by means of hardware devices where no data are stored. On the other hand, the technologies used are innovative but mature enough to avoid any security risk. Specifically, the FIWARE framework [28] is considered in order to use the MQTT protocol as a message broker through the Mosquitto implementation and MongoDB databases and servers that meet all security requirements according to European standards.

### 3.4. Cloud-Based Data Storage and Analysis Infrastructure

To store and analyse the collected data, an Ubuntu server virtual machine was deployed in a cloud infrastructure. In this machine, a MongoDB database was created containing four collections, namely *ssidCollect*, *crowdLevel*, *places*, and *dataSSIDCollector*. The purpose of these collections are:

- *ssidcollect*: contains the data coming from each sensor of the HOPU Smart Spot devices. Specifically, this collection contains the following attributes:
  - *sensor* (string): stores the ID of the sensor from which it has obtained the data.
  - *date* (timestamp): the exact date on which the information was captured.
  - *mac* (SHA1): information obtained by the sensor, specifically the MAC of the device after applying the SHA1 algorithm.
  - *ssid* (SHA1): information obtained by the sensor, specifically the SSID of the PNL from the corresponding MAC, or rather the summary function of that SSID after applying the SHA1 algorithm.
- *crowdLevel*: stores the number of people in a place at a given time. The following attributes are stored in this collection:
  - *sensor* (string): stores the ID of the sensor that has obtained the data.
  - *date* (timestamp): the exact date on which the information has been captured.
  - *crowd*: stores the number of people detected by the sensor nearby.
- *places*: stores information about the place where the sensors are installed. Specifically, this collection contains the following attributes:
  - *sensor* (string): stores the ID of the sensor that has obtained the information.
  - *name* (string): name of the place where the sensor is located.
  - *description* (string): a short description of the sensor location.
  - *coords* (coordinates): coordinates of the sensor location.
  - *city* (string): name of the city where the sensor is located.
- *dataSSIDCollector*: contains the SSID data collected by the device. Attributes are as follows:
  - *date* (timestamp): the exact date on which the data have been captured.
  - *ssid* (SHA1): information obtained by the device, specifically the SSID, or rather the summary function of that SSID after applying the SHA1 algorithm.
  - *location* (string): name of the city where the data was obtained.

Additionally, the *summaryInfo* collection is created to store daily aggregated data. This collection has the summary of the data captured by the sensor for each day (including the number of visitors and residents). This collection contains the following attributes:

- *sensor* (string): stores the ID of the sensor that has obtained the information.
- *date* (timestamp): day for which the summary information is displayed.
- *nVisitors* (integer): number of visitors detected.
- *nResidents* (integer): number of residents detected.

○    *total* (integer): total number of people detected.

The *summaryInfo* collection is filled by applying a consolidation algorithm, whose pseudocode is shown below in Listing 1. This algorithm uses the collected data to obtain a daily summary of the number of visitors and residents detected by a specific sensor.

**Listing 1.** Consolidation algorithm to obtain a daily summary of the number of visitors and residents detected by a specific sensor.

```
SSIDRatio = 0.5;
frequentMAC = 50;
for each sensor in available_sensors
{
    //Set initial values
    total = 0;
    visitors = 0;
    residents = 0;

//Get MACs and number of occurrences, associated SSIDs and SSIDs
that match our database

macs = getMACInfo(MAC);

    for each MAC detected as input_mac
    {
        //Number of MAC appearances
        count = inpute_mac.count;

        //Check if it is not a usual MAC of the place
        if (count < frequentMAC)
        {
            //Number of associated SSIDs
            totalSSIDs = input_mac.ssids.length;

            //Number of associated local SSIDs
            localSSIDs = input_mac.ssidsMatch.length;

            //Percentage of local SSIDs ratio
            ratio = localSSIDs / total SSIDs;

            total++;

            // If the proportion of local SSIDs is greater than
        0.50, we consider that this MAC belongs to a
        resident, otherwise she/he is a visitor

            if (ratio > SSIDRatio)
               resident += 1;
            else
               visitors += 1;

        }

        //Save on the DB
        insertDB (timestamp,sensor,total,residents,visitors);
    }
}
```

The *SSIDRatio* is the percentage of preferred SSIDs of a device that must belong to the city to be considered as a device owned by a resident. The default value is 0.5. On the other

side, *frequentMAC* is the number of times that a MAC is detected each day for the same sensor in order to be counted as a unique device (e.g., because it belongs to a worker). The default value is 50. Finally, *list_frequent_SSIDs* is a list whose elements are hashed SSIDs. They come from a data source with common SSID names that cannot be classified. Data come from [27] and the reason for this use is explained later.

In the virtual machine, the data are processed in order to differentiate visitors from residents. To do so, the HOPU Smart Spot devices obtain the preferred SSID data from the mobile devices within range. The HOPU Smart Spot device sends the SSID along with the date and time, as well as its identification (location) to a broker developed in Mosquitto (which implements the MQTT protocol for message management). This broker is fundamental to this system, as it allows the management of context information, querying and updating it (context being understood as the sensors that produce the data).

When the broker receives a message, it notifies a backend built in Node.js to process the message information and store it in the MongoDB database. In addition, this backend, at the end of each day, consolidates the collected information and stores it in MongoDB as well.

To access the consolidated daily data, an API has been implemented to provide data about the locations where the sensors are located. This API can be accessed by developers who have permission to consult its information (the permission is provided through an API key).

A web API has been developed in order to consume data. Specifically, there are two main functions. The first one is */places* which returns the list of all the locations where sensors are placed, including a description, coordinates, and city, as well as a sensor ID. Figure 3 (left) shows an excerpt of an output of the /places containing two sensors in the pilot study of Alcoi (Spain): one of them "fontRoja" located in a natural park and the other one "touristInfo" located in the Tourist Information Office both in the city of Alcoi (Spain).

```
[
  {
    "name": "Parque natural de la Font Roja",
    "desc": "El Parque Natural de la Font Roja, se encuen
    "coords": {
      "lat": 38.6644071,
      "lng": -0.540113
    },
    "sensor": "fontRoja",
    "city": "Alcoi"
  },
  {
    "name": "Tourist-Info Alcoy",
    "desc": "Centro de información turística de Alcoy",
    "coords": {
      "lat": 38.6980065,
      "lng": -0.4729549
    },
    "sensor": "touristInfo",
    "city": "Alcoi"
  }
]
```

```
[
  {
    "total": 2,
    "nVisitors": 2,
    "nResidents": 0,
    "date": 1606345200,
    "sensor": "touristInfo"
  }
]
```

**Figure 3.** Excerpt example for */places* (**left**), and for */summaryInfo* (**right**).

Secondly, there is the */summaryInfo* function that returns the daily report for a specific sensor (identified by its sensor ID) containing the total number of people for a specific date, as well as the number of them being visitors and residents. The parameters are as follows:

- *sensor*: ID of the sensor from which we want to get the data.
- *day*: date from which we want to get the data.
- If we want the data in a date range, we may indicate the date on which the range begins in the start parameter and the date on which the range ends in the end parameter.

Figure 3 (right) shows an excerpt of an output of the */summaryInfo* containing data from a specific day obtained by the "touristInfo" sensor.

## 4. Results

As mentioned above, a pilot study is being carried out in the city of Alcoi (Spain). This is a medium-sized city (see Figure 4) with a population of 58,994 inhabitants (2019), most of whom live in the urban area. The architecture of the urban area includes mainly 3- and 4-storey buildings.

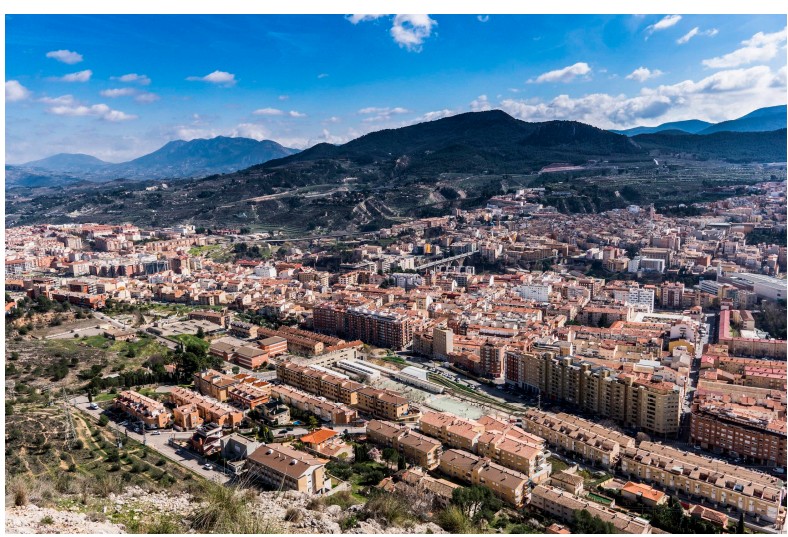

**Figure 4.** View of the city of Alcoi (Spain) (source: Jordi Miró).

Alcoi is a city of long industrial tradition in a process of economic diversification in which visitor economy is playing an emergent role thanks to its rich heritage, both natural (two natural parks located in the municipal area) and cultural (a World Heritage site among other singular attractions of different historical periods). Despite the drawback of a scarce accommodation supply (585 beds), the number of visitors shows the steady growth and first signals of congestion associated with cultural events and tourist hotspots. According to this trend, the local DMO, highly committed to smart destination initiatives, is trying to develop new crowd monitoring systems, a goal reinforced by the need to guarantee social distancing in order to assure a safe visitor experience in the COVID-19 context. Specifically, the Alcoi DMO needs to distinguish visitors from residents on a daily basis at several spots in the city by avoiding shortcomings of traditional burdensome surveys or other intrusive approaches such as a destination app, while privacy is preserved.

To make an initial assessment of the collection of SSIDs from the city, one vehicle and the collection system were used in a very small portion of the city for 60 min. The SSIDs were collected unencrypted and timestamped.

A total of 7184 unique SSIDs were captured. To contrast the captured SSIDs with the values stored in geolocation and wardriving capture services, the database was compared with the WiGLE database for the area of interest. Figure 5 shows the WiGLE map of SSIDs for the urban area of Alcoi. The purple dots on the map indicate where a particular hotspot has been detected by the WiGLE mobile application. The position of the hotspot is estimated from the GPS position of the mobile phones that have the application installed and detect the hotspot.

From the comparison with WiGLE's SSIDs, it is concluded that SSID names are very dynamic, changing continuously over time. This would require periodic refreshing of the local SSID database.

Another conclusion from this first analysis is that there are very common names worldwide that should be discarded in the scan, for example, the names: *AndroidAP*, *ASUS*, *dlink* or *hpsetup*. These should probably be discarded both for the local SSID database and the HOPU Smart Spot. The database of most common SSIDs collected by WiGLE would be useful to facilitate this filtering.

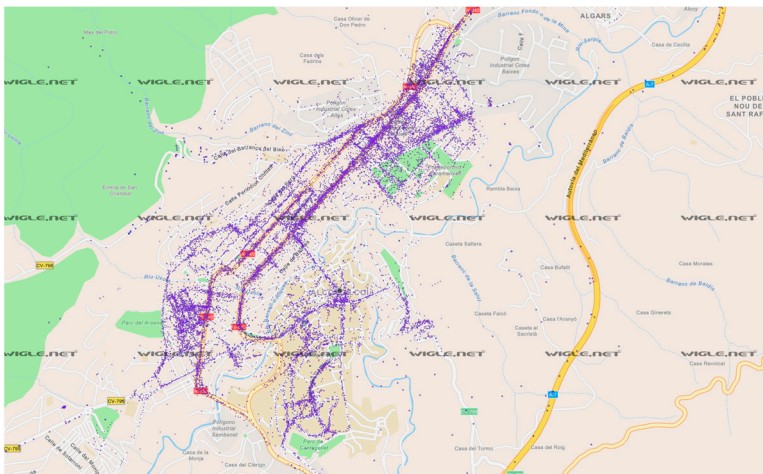

**Figure 5.** WiGLE map of SSIDs collected in the city of Alcoi.

Another curious conclusion is that the major phone operators will repeat SSIDs of their WiFi router in many places. For example, "MOVISTAR_1DBE" (hexadecimal string modified for privacy) only allows $2^{16}$ = 65,536 combinations and, therefore, will be present in more than one city. In any case, we consider it valid to discriminate, accepting that it compensates for the system getting some SSIDs wrong. Some router vendors have updated this issue by using longer combinations, e.g., "TP-LINK_44E4C8" (hexadecimal string modified for privacy).

Then, in order to validate our approach, it was launched in Alcoi for 14 days (from 1 March 2021 to 14 March 2021) for 2 h each day (from 11:00 h to 13:00 h). The rationale behind this timetable is that a dedicated person from Alcoi's Tourist Info office was involved in manually conducting surveys with people that came in. Each survey collects the following information:

- Question 1. Timestamp when information was collected. This question allows us to know when the citizen was in Alcoi.
- Question 2. Town of residence. This question allows us to determine if the respondent is a resident of Alcoli or a visitor from another town.
- Questions 3 and 4. How many people do you travel with? How many of these people you travel with have entered the Tourist Info office? These two questions allow us to know additional information about respondents of the survey, including if they currently have more than one WiFi device.
- Question 5. Do you have the WiFi on your mobile phone enabled to connect to available WiFi networks? This question is useful for empirically determining the percentage of individuals that are using WiFi in order to better estimate the number of visitors, thus allowing our approach to fit real-world data.

Figure 6 shows a flowchart stating how the results of our validation approach are collected. Table 1 shows the results of crowd monitoring using the HOPU Smart Spot located in the Alcoi Tourist Info office with regard to the data collected from the surveys in the same location (Table 2 summarizes results, including average accuracy). It can be observed that the total number of people detected (visitors and residents) is often overestimated with our approach. However, we have to analyse specific results detecting visitors versus residents. According to the flowchart, we compare the values coming from the detection of visitors in our proposal with those coming from the surveys carried out daily in the tourist offices. The result of this comparison can be seen in Figures 7–9, where the calculation of the average accuracy is also included.

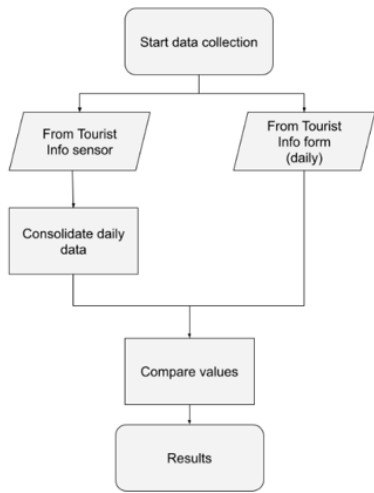

**Figure 6.** Flowchart representing how results are collected from our validation approach.

**Table 1.** Crowd monitoring results by using HOPU Smart Spots vs. data collected by surveys.

| | HOPU Smart Spot | | | Survey | | |
|---|---|---|---|---|---|---|
| Date | Visitors | Residents | Total | Visitors | Residents | Total |
| 1 March 2021 | 5 | 1 | 6 | 1 | 1 | 2 |
| 2 March 2021 | 1 | 1 | 2 | 1 | 1 | 2 |
| 3 March 2021 | 4 | 2 | 6 | 2 | 3 | 5 |
| 4 March 2021 | 4 | 0 | 4 | 3 | 1 | 4 |
| 5 March 2021 | 0 | 0 | 0 | 0 | 0 | 0 |
| 6 March 2021 | 11 | 2 | 13 | 9 | 1 | 10 |
| 7 March 2021 | 9 | 1 | 10 | 8 | 2 | 10 |
| 8 March 2021 | 10 | 4 | 14 | 1 | 4 | 5 |
| 9 March 2021 | 6 | 1 | 7 | 4 | 2 | 6 |
| 10 March 2021 | 7 | 2 | 9 | 4 | 2 | 6 |
| 11 March 2021 | 8 | 3 | 11 | 5 | 1 | 6 |
| 12 March 2021 | 7 | 1 | 8 | 6 | 2 | 8 |
| 13 March 2021 | 19 | 3 | 22 | 17 | 4 | 21 |
| 14 March 2021 | 2 | 3 | 5 | 2 | 0 | 2 |
| 15 March 2021 | 4 | 2 | 6 | 4 | 1 | 5 |
| 16 March 2021 | 3 | 3 | 6 | 4 | 0 | 4 |
| 17 March 2021 | 6 | 5 | 11 | 6 | 2 | 8 |
| 18 March 2021 | 7 | 3 | 10 | 6 | 2 | 8 |
| 19 March 2021 | 8 | 1 | 9 | 7 | 3 | 10 |
| 20 March 2021 | 5 | 2 | 7 | 6 | 2 | 8 |
| 21 March 2021 | 4 | 4 | 8 | 4 | 2 | 6 |
| 22 March 2021 | 5 | 3 | 8 | 5 | 0 | 5 |
| 23 March 2021 | 5 | 4 | 9 | 4 | 3 | 7 |
| 24 March 2021 | 4 | 3 | 7 | 4 | 3 | 7 |
| 25 March 2021 | 2 | 1 | 3 | 2 | 3 | 5 |
| 26 March 2021 | 11 | 1 | 12 | 14 | 2 | 16 |
| 27 March 2021 | 6 | 4 | 10 | 6 | 2 | 8 |
| 28 March 2021 | 5 | 4 | 9 | 4 | 3 | 7 |
| 29 March 2021 | 2 | 3 | 5 | 2 | 2 | 4 |
| 30 March 2021 | 3 | 2 | 5 | 4 | 2 | 6 |
| 31 March 2021 | 2 | 1 | 3 | 1 | 1 | 2 |

**Table 2.** Summary of results, including average accuracy.

| | |
|---|---|
| Detected Visitors ($V_d$) | 175 |
| Actual visitors ($V_a$) | 146 |
| Average accuracy: ($V_d - V_a$) * 100/$V_d$ | 83% |

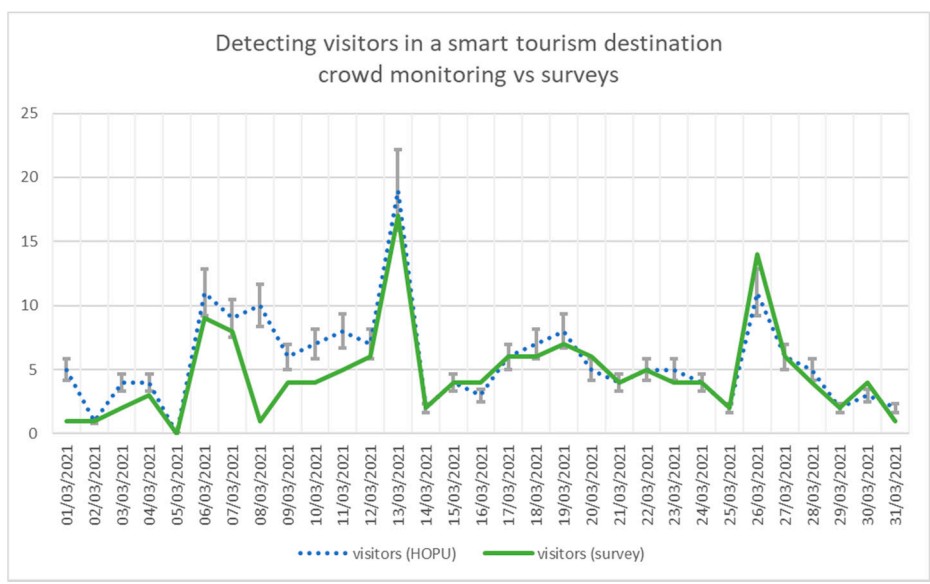

**Figure 7.** Using HOPU Smart Spot vs. manually conducted surveys to determine number of visitors in Alcoi.

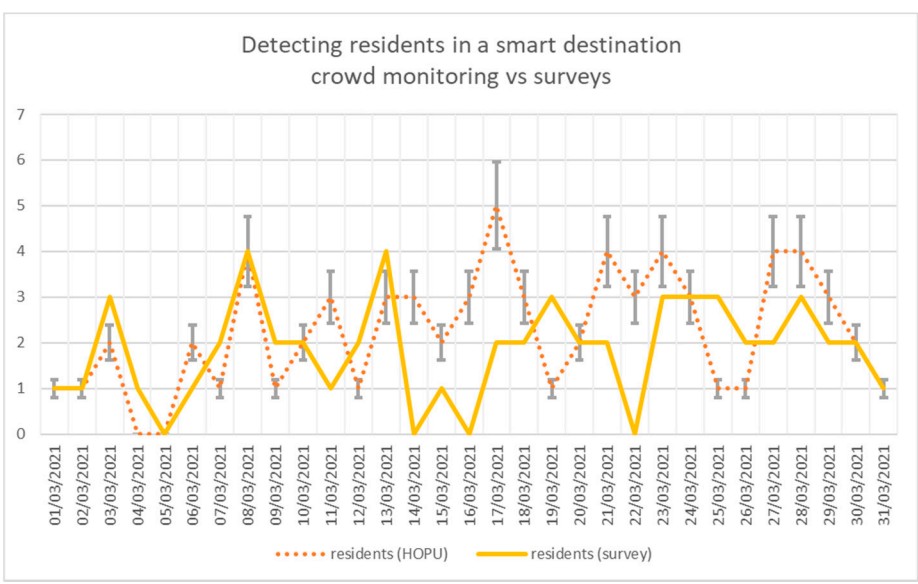

**Figure 8.** Using HOPU Smart Spot vs. manually conducted surveys to determine number of residents in Alcoi.

Figure 7 shows the number of visitors detected by using our approach compared to the results of the surveys. Although the number of visitors is overestimated (except from 02/03, 05/03 and 14/03), the trend is shown to be equivalent. The only day the trend is broken is 08/03. Additionally, it can be observed that Saturdays (06/03 and 13/03) are the days on which more visitors are detected (which is an obvious result for a tourist destination).

Figure 8 shows the number of residents detected by using our approach compared to the results of the surveys. The number of residents is often underestimated by our

approach. On several days, the numbers of residents detected are the same when using both mechanisms (01/03, 02/03, 05/03, 08/03, and 10/03), while on two days (06/03, and 11/03), the number of residents is overestimated when using our approach.

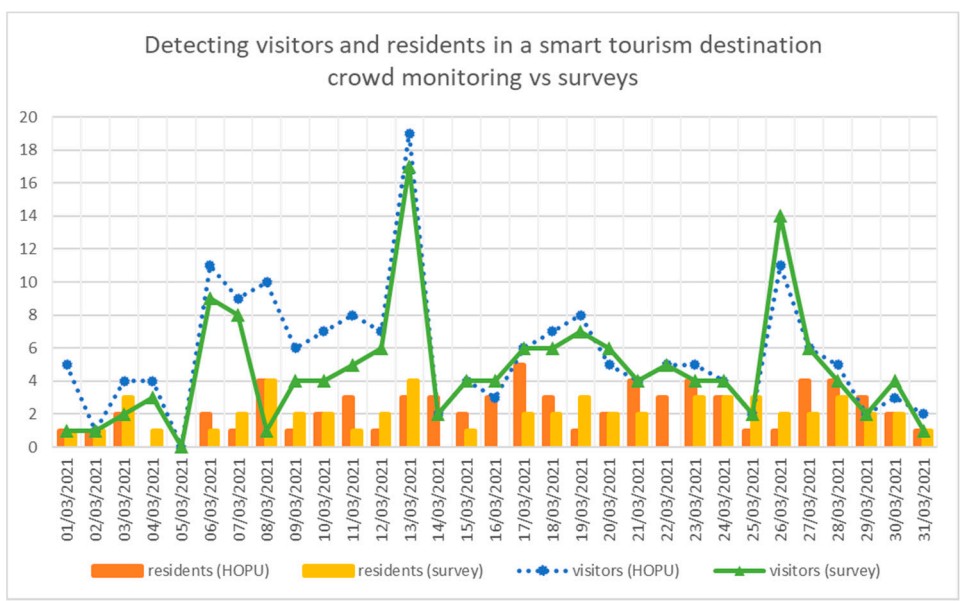

**Figure 9.** Comparing crowd monitoring visitors and residents in Alcoi by using HOPU Smart Spot vs. manually conducted surveys.

Figure 9 compares the number of visitors and the number of residents detected using both approaches (HOPU Smart Spots and surveys).

## 5. Discussion

Initial results show that the processing of preferred SSIDs emitted by mobile phones is useful for distinguishing residents from visitors. According to Table 1 and Figures 7–9, the results are promising. Table 1 shows that two days (02/03, and 05/03) present correct results. On several days (specifically 01/03, 06/03, 08/03, 10/03, 11/03, and 14/03), more people were detected with our approach based on HOPU Smart Spot than people that actually visited the Tourist Info office (according to the manually conducted survey). This is mainly caused by the WiFi scanning range of the HOPU Smart Spot. Additionally, on several days (03/03, 04/03, 07/03, 09/03, 12/03, and 13/03), residents were detected as visitors. This is mainly because our list of SSIDs belonging to Alcoi's WiFi devices is incomplete. Our approach never underestimates the total number of persons detected, as we asked the people that answered the survey in the Tourist Info office whether their WiFi on their mobile phone was enabled or not. People with a disabled WiFi connection are not considered. On the other hand, the total number of people detected is greater for our approach based on HOPU Smart Spots than for manually conducted surveys, mainly due to the need to better adjust the operating range of the device (i.e., if the operating range is too wide, then mobile phones from outside the Tourist Info office could be detected).

## 6. Conclusions and Future Work

Our approach to crowd monitoring is based on the WiFi scanning of mobile phones. Thus, it is a suitable tool to be used by the DMO from cities such as Alcoi to prevent crowds in the pandemic times we are living in, by distinguishing visitors and residents on a daily basis at several spots in the city, while privacy is preserved. This enables Alcoi to avoid the shortcomings of traditional burdensome surveys or other intrusive approaches such as a destination app. Moreover, it is an initial step to further consider tourist digital footprints or data traces from tourist activities, as they occur if a person can be considered a visitor,

as stated in [13]. Remarkably, in our approach, data analysis is performed at the end of each day in order to be very privacy conscious. Within the margins offered by the GDPR, it would be possible in the future to perform these analyses in real time to improve the responsiveness of a smart destination.

It is worth recalling that our approach is GDPR-compliant, since we anonymize directly after receiving the packet, having a hash and a salt to avoid people being identified. (Actually, re-identification is only possible for the duration of a not-changed salt.) Consequently, anonymization is ensured while having the role of a "maintainer" of the device. Since we do not store or process private data, we do not need the consent of individuals, and we only release anonymised data from WiFi scanning devices. Additionally, as the city owns and operates the system, it is not the individual companies that provide the technical guarantees mentioned above, but it is the system operator (usually the city) that complies with the GDPR.

Promising results from our pilot study allow us to envision interesting future work. First, our approach should be linked to other big data sources that are already being used in smart destinations [13] (such as data from social media, booking services, destination cards and passive mobile data) in order to deploy an effective DMO. For example, in the particular case of Alcoi, techniques such as surveys in tourist offices, vehicle access to the city through number plate analysis, etc., are used and they can be considered to complement our approach. Additionally, extending the technique of SSID collection to other nearby cities would allow a better understanding of the flow between adjacent towns, leading to a better classification of visitors and, potentially, to a larger-scale DMO tool. Finally, we are currently improving the implementation of our approach to automate tasks that require human intervention and are therefore costly in economic terms. On the one hand, it is essential that the destination takes over the collection tasks of SSIDs without incurring an additional cost; in that sense, we have decided to use the waste collection service to install our IoT device SSID collector. On the other hand, the collection of SSIDs should be as quick as possible, and in that sense, we are developing a collector that does not need human intervention and automatically downloads the SSID data when detecting specific WiFi points of the municipal services.

Finally, we would like to highlight that this approach could be applied to other contexts within a smart tourism destination, e.g., organization of a congress in order to know the behaviour of attendants or sporting events in which DMOs want to know the benefits brought to the hospitality sector in the destination.

**Author Contributions:** Conceptualization, J.A.I.-B., A.J.J., J.-N.M. and A.P.; methodology, J.A.I.-B., A.J.J., J.-N.M. and A.P.; hardware and software, A.B., D.F.R., A.G.-O. and J.L.; pilot study, A.B., D.F.R., A.G.-O., J.A.I.-B. and A.J.J; validation, A.B., J.-N.M. and A.P.; writing—original draft preparation, A.B., D.F.R., A.G.-O., J.A.I.-B., A.J.J., J.L., J.-N.M. and A.P. Authors are alphabetically listed in the paper. All authors have read and agreed to the published version of the manuscript.

**Funding:** This research was carried out within the research Project Alcoi Tourist Lab framework, co-funded by the Alcoi City Council & the Valencian Innovation Agency. The research was also partially funded by project UAPOSTCOVID19-10 from the University of Alicante. Finally, this research was partly supported by the EU CEF project GreenMov, CARM HORECOV-21 project (https://horecovid.com/ (accessed on 12 January 2022)). is financed through the Call for Public Aid destined to finance the Strategic projects contemplated in the Research and Innovation Strategy for Smart Specialization - RIS3MUR Strategy by the Autonomous Community of the Region of Murcia, through the Ministry of Economic Development, Tourism and Employment within the framework of the FEDER Region of Murcia Operational Program 2014–2020 within the framework Thematic Objective 1. Strengthen research, technological development and innovation by 80% and with CARM's own funds in 20%, and finally the EU project H2020 NIoVE (833742).

**Acknowledgments:** We would like to thank Paola Pons and Pedro Ramiro from the Alcoi City Council, involved in the Project Alcoi Tourist Lab. Additionally, we would like to thank the anonymous referees for all their comments and suggestions that allow us to improve the manuscript.

**Conflicts of Interest:** The authors declare no conflict of interest.

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
