# Peer review of "Crowd Monitoring in Smart Destinations Based on GDPR-Ready Opportunistic RF Scanning and Classification of WiFi Devices to Identify and Classify Visitors’ Origins"

_electronics, doi:10.3390/electronics11060835_

Round 1

Reviewer 1 Report

The article presents a crowdsensing approach for popular destinations using WiFi technology to classify visitors from residents. This is an approach to make destinations safer in the current pandemic concern. The approach analyses WiFi probe requests to identify whether the received request comes from a visitor smart device or not without any privacy concern. The article is interesting and well written, however, the below comments should be addressed before publication.

  • In the abstract: state the efficiency of the approach in numbers e.g. how accurate is the approach in detecting visitors?
  • State the full acronym in every first occurrence.
  • Discuss an approach to estimate the number of visitors that are not using WiFi so the proposed model can fit real-world data.
  • How does your approach account for users using multiple WiFi devices? Discuss in the paper.
  • The algorithm does not have a title. Complement the algorithm with in-line comments.
  • The main concern that arises is regarding Table 1. How one can conclude with such a small number of surveys? This can lead to an accurate result.
  • Include a confidence interval (e.g., 95%) in all your charts.
  • Provide a reference of what is a Wigle map.
  • Future works should be stated clearly in a separate paragraph to facilitate the readers to find them.

Overall, the article proposes a new approach that estimates the number of visitors to popular destinations. However, my main concern is how conclusions are drawn with such a small number of surveys? Results can be inaccurate. The paper is well organised and technically sound. Authors are encouraged to address the above suggestions.

Author Response

Responses to reviewer 1

Reviewer #1, Comment #1

The article presents a crowdsensing approach for popular destinations using WiFi technology to classify visitors from residents. This is an approach to make destinations safer in the current pandemic concern. The approach analyses WiFi probe requests to identify whether the received request comes from a visitor smart device or not without any privacy concern. The article is interesting and well written, however, the below comments should be addressed before publication.

Authors response: We would like to thank the reviewer for this comment, and we will do our best to address the remainder comments, which will allow us to improve the document.

Reviewer #1, Comment #2

In the abstract: state the efficiency of the approach in numbers e.g. how accurate is the approach in detecting visitors?

Authors response: We would like to thank reviewer for this comment. In order to address this comment, we have included accuracy of our approach in the abstract.

Reviewer #1, Comment #3

State the full acronym in every first occurrence.

Authors response: We have improved the paper in accordance with this suggestion.

Reviewer #1, Comment #4

Discuss an approach to estimate the number of visitors that are not using WiFi so the proposed model can fit real-world data.

Authors response: We are grateful for this suggestion to improve the paper. We have tried to clarify that the proposed method does not require the mobile to be connected to the destination's WiFi, it simply requires the possibility to connect to it to be enabled. For example, any mobile phone that automatically connects to the tourist's home or hotel WiFi will be captured by this system.

Also, our survey included a question to state if visitors are using WiFi or not (question number 5: “Do you have the WiFi on your mobile phone enabled to connect to available WiFi networks?”), but we have noticed that we did not mention that we used this empirical information in our approach for dealing with visitors not using WiFi. We have explained it in more detail when the conducted survey is described in the paper, as follows:

  • “Question 5: Do you have the WiFi on your mobile phone enabled to connect to available WiFi networks? This question is useful for empirically determining percentage of individuals that are using WiFi in order to better estimate the number of visitors, thus allowing our approach to fit real-world data.”

Reviewer #1, Comment #5

How does your approach account for users using multiple WiFi devices? Discuss in the paper.

Authors response: We are grateful for this suggestion to improve the paper. Our survey included a question to ask for the number of devices per person (question numbers 3 & 4: “How many people do you travel with? How many of these people you travel with have entered the Tourist Info office?”), but we have noticed that we did not mention that we used this empirical information in our approach. We have explained it in more detail when the conducted survey is described in the paper, as follows:

  • “Questions 3 and 4: How many people do you travel with? How many of these people you travel with have entered the Tourist Info office? These two questions allow us to know additional information about respondents of the survey, including if they currently have more than one WiFi device.”

Reviewer #1, Comment #6

The algorithm does not have a title. Complement the algorithm with in-line comments.

Authors response: We have updated the algorithm consequently, i.e., he added a caption as well as in-line comments.

Reviewer #1, Comment #7

The main concern that arises is regarding Table 1. How one can conclude with such a small number of surveys? This can lead to an accurate result.

Authors response: Unfortunately, conducting manual surveys is costly and, particularly in this case, has been a challenge due to the pandemic. We have included information from the whole month of March not previously available when we submitted the manuscript. Therefore, we doubled the number of surveys. We have updated table with the information and figures, accordingly, in the manuscript.

Reviewer #1, Comment #8

Include a confidence interval (e.g., 95%) in all your charts.

Authors response: Done.

Reviewer #1, Comment #9

Provide a reference of what is a Wigle map.

Authors response: Thank you very much for detecting this issue. We explained what is WiGLE including, this text and other minor corrections.

WiGLE (or Wireless Geographic Logging Engine) is a website for collecting information about the different wireless hotspots around the world. Users can register on the website and upload automatically hotspot data like GPS coordinates, SSID, MAC address and the encryption type used on the hotspots discovered using a mobile application. Considered that this information is open, we decided to use it as a reference to compare some of the data collected by WiGLE and the one collected by our system.

Reviewer #1, Comment #10

Future works should be stated clearly in a separate paragraph to facilitate the readers to find them.

Authors response: Done. We have included a new section 6, entitled “Conclusions and future work” to briefly give some conclusions of our work as well as to sketch out future works.

Reviewer #1, Comment #11

Overall, the article proposes a new approach that estimates the number of visitors to popular destinations. However, my main concern is how conclusions are drawn with such a small number of surveys? Results can be inaccurate. The paper is well organised and technically sound. Authors are encouraged to address the above suggestions.

Authors response: We would like to thank the referee for his/her comments. We improved the paper in accordance with all your suggestions and those of the other referees.

Reviewer 2 Report

Dear Author(s), I have reviewed the paper, the following are the review comments- 

  1. References [18, 19, 20, 21, 22] are not being cited in the manuscript.
  2. Discuss fig. 5 in the manuscript.
  3. How authors simulated figure 6, 7, and 8, not clear.
  4. Authors must show either flow chart or algorithm in order to justify their simulation results.
  5. ,Picture quality of fig 1 & 2 must be improved.   

Author Response

Reviewer #2, Comment #1

Dear Author(s), I have reviewed the paper, the following are the review comments-

Authors response: We would like to thank the referee for his/her comments. We have attempted to improve the paper in accordance with all of the referee’s suggestions.

Reviewer #2, Comment #2

References [18, 19, 20, 21, 22] are not being cited in the manuscript.

Authors response: Please, accept our apologies for this mistake, since we have a misunderstanding with different versions of the related work section. Now, in the new version of the manuscript, we have included a description of these references, as follows:

“Lately, novel approaches have emerged for detecting people and understanding their behaviour automatically [19]. For example, approaches (such as [18]) attempt to transform an unmodified WiFi radio infrastructure into a flexible sensing system for detecting the people moving indoors. Other proposals, such as [22], develop their own sensor nodes to count the number of pedestrians, their direction of travel along with some ambient parameters. Interestingly, proposals such as [20], use stereo thermal camera setup for pedestrian counting and behaviour understanding regardless light conditions. Finally, unexpected COVID-19 pandemic scenario has encouraged the development of systems monitor achievement of social distancing. In this sense, [21] proposes using low-cost Raspberry Pi and a variety of sensors to measure temperature, distances, etc. in order to ensure the COVID-19 standard operating procedure compliance.”

Reviewer #2, Comment #3

Discuss fig. 5 in the manuscript.

Authors response: Thank you very much for detecting this issue. We explained what is shown in that figure and also include extra text describing WiGLE. See both fragments below.

“… Figure 5 shows the WiGLE map of SSIDs for the urban area of Alcoi. The purple dots on the map indicate where a particular hotspot has been detected by the WiGLE mobile application. The position of the hotspot is estimated from the GPS position of the mobile phones that have the application installed and detect the hotspot.”

“… WiGLE (or Wireless Geographic Logging Engine) is a website for collecting information about the different wireless hotspots around the world. Users can register on the website and upload automatically hotspot data like GPS coordinates, SSID, MAC address and the encryption type used on the hotspots discovered using a mobile application. Considered that this information is open, we decided to use it as a reference to compare some of the data collected by WiGLE and the one collected by our system.”

Reviewer #2, Comment #4

How authors simulated figure 6, 7, and 8, not clear. Authors must show either flow chart or algorithm in order to justify their simulation results.

Authors response: We have added a flow chart (new figure 6) to show how we get our results; this flow chart complements the algorithm now defined as listing 1.

According to the flowchart we have added, we compare the values coming from the detection of visitors coming from our proposal with those coming from the surveys carried out daily in the tourist offices. The result of this comparison can be seen in figures 7, 8 and 9, where the calculation of the average accuracy is also included. This explanation has been added to the paper.

Reviewer #2, Comment #5

Picture quality of fig 1 & 2 must be improved.  

Authors response: Done (better quality versions of the figures have been included).

Reviewer 3 Report

The presented methodology for the monitoring the tourist behavior based on wifi networks using the preferred SSID of the mobile phones raises serious privacy issues (already mentioned by the authors). The proposed approach is based only on data collected through wifi spots but we know that the majority of the tourists access the internet through the local mobile providers (roaming). As a result the collected data do not present the current situation and the conclusions is rather useless. The technical part of the methodology is interesting and useful, but in our opinion, it should be applied in other contexts e.g. monitoring the participants of a congress (in that case GDPR restrictions can be alleviated).

Author Response

Reviewer #3, Comment #1

The presented methodology for the monitoring the tourist behavior based on wifi networks using the preferred SSID of the mobile phones raises serious privacy issues (already mentioned by the authors).

Authors response: We would like to thank the reviewer very much for this concern, in fact, it was precisely our main concern before launching this initiative. Before initiating it, the proposal went through the legal advisory service of the city council specialised in data protection at European level. Of course, the system has a lot more potential that we cannot exploit, but what is presented in the article is fully GDPR compliant. In this sense, we would like to point out two key aspects of the solution that helped us to comply with GDPR:

  • The city owns and operates the system. Therefore, it is not the individual companies that provide the technical guarantees, but it is the system operator (usually the city) that complies with the GDPR, which in Article 6 of the GDPR grants access to this information for security and safety purposes, such as crowd management, as well as declaring that it is not about handling private data.
  • We anonymize directly after receiving the packet. We had a hash and a salt, so that people cannot be identified, and re-identification is only possible for the duration of a not changed salt. Therefore, after this action, the data collected is no longer private, and we are GDPR-compliant after that. Consequently, in the case of our solution, we ensure anonymization while having the role of a “maintainer” of the device. Since we do not store or process private data, we do not need the consent of individuals, and we only release anonymised data from WiFi scanning devices.

In order to improve our manuscript, we have included a new paragraph in the conclusions, as follows:

“It is worth to recall that our approach is GDPR-compliant, since we anonymize directly after receiving the packet, having a hash and a salt to avoid people to be identified (actually, re-identification is only possible for the duration of a not changed salt). Consequently, anonymization is ensured while having the role of a “maintainer” of the device. Since we do not store or process private data, we do not need the consent of individuals, and we only release anonymised data from WiFi scanning devices. Also, as city owns and operates the system, it is not the individual companies that provide the technical guarantees mentioned above, but it is the system operator (usually the city) that complies with the GDPR.”

Reviewer #3, Comment #2

The proposed approach is based only on data collected through wifi spots but we know that the majority of the tourists access the internet through the local mobile providers (roaming). As a result the collected data do not present the current situation and the conclusions is rather useless.

Authors response: We would like to thank the reviewer for this useful comment, which made us aware that a more detailed explanation of why our approach uses data from WiFi-enabled spots (nor real WiFi hotspot) is required in order to better understand usefulness of our approach.

We have tried to clarify that the proposed method does not require the mobile to be connected to the destination's WiFi, it simply requires the possibility to connect to it to be enabled. For example, any mobile phone that automatically connects to the tourist's home or hotel WiFi will be captured by this system.

Also, managers of tourism destinations (i.e., Destination Management Organisation - DMO) require crowd monitoring data for supporting decision making. First approaches of many DMOs were focused on buying data from mobile providers, but they depend on third parties, as well as budget availability. Therefore, relying only on data from mobile providers is not a realistic and sustainable solution and other more independent solution (as our approach) that provides a better data governance are required. Consequently, we have included a better explanation of these points in the introduction as follows:

“At this point it should be noted that our approach complements other types of initiatives from DMOs based on the purchase of data from third parties (mobile providers). In fact, many DMOs have based their decision-making process on data acquisition from mobile providers but this attempts against independence since it is difficult to ensure high-standard data governance principles. Moreover, relying solely on third party data means that DMOs are constrained by budgetary availability.”

Reviewer #3, Comment #3

The technical part of the methodology is interesting and useful, but in our opinion, it should be applied in other contexts e.g. monitoring the participants of a congress (in that case GDPR restrictions can be alleviated).

Authors response: We would like to thank reviewer for this comment. We fully agree that other scenarios would be more appropriate for this solution and we are glad you proposed them, good idea. In our case, the problem was the other way around, we wanted to monitor visitors to a city and we came up with several “legal” approaches to do so and this is one of them. Consequently, we have included new explanations about application of our approach in other contexts in the “6. Conclusions and future work”, as follows:

“Finally, we would like to highlight that this approach could be applied to other contexts within a smart tourism destination, e.g., organization of a congress in order to know the behaviour of attendants or sporting events in which DMOs want to know benefits brought to the hospitality sector in the destination.”

Round 2

Reviewer 1 Report

I would like to thank the authors for putting effort into addressing all my suggestions.

I am satisfied with all reponses.

Thanks.

Reviewer 3 Report

The authors properly revised the manuscript according to reviewers' comments and in the present form it is suitable for publication.